# Design and Simulation Study of an Optical Mode-Localized MEMS Accelerometer

**DOI:** 10.3390/mi14010039

**Published:** 2022-12-23

**Authors:** Yu Feng, Wuhao Yang, Xudong Zou

**Affiliations:** 1Key Laboratory of Low Altitude Monitoring Network Technology, QiLu Aerospace Information Research Institute, Jinan 250101, China; 2State Key Laboratory of Transducer Technology, Aerospace Information Research Institute, Chinese Academy of Sciences, Beijing 100190, China; 3School of Electronic, Electrical and Communication Engineering, University of Chinese Academy of Sciences, Beijing 100049, China

**Keywords:** suspended directional coupler, optical mode localization, micro-opto-electro-mechanical system, displacement detection

## Abstract

In this paper, we demonstrate a novel photonic integrated accelerometer based on the optical mode localization sensing mechanism, which is designed on an SOI wafer with a device layer thickness of 220 nm. High sensitivity and large measurement range can be achieved by integrating coupled ring resonators with a suspended directional coupler on a proof mass. With the help of FEA simulation and numerical analysis, the proposed optical mode-localized sensor presents a sensitivity of 10/g (modal power ratio/acceleration) and an inertial displacement of from −8 to 10 microns corresponding to a range from −23.5 to 29.4 g. The free spectral range is 4.05 nm around 1.55 microns. The acceleration resolution limited by thermomechanical noise is 4.874 μg. The comprehensive performance of this design is competitive with existing MEMS mode localized accelerometers. It demonstrates the potential of the optical mode-localized inertial sensors as candidates for state-of-the-art sensors in the future.

## 1. Introduction

As a basic sensing unit for acceleration, velocity, and position measurement, accelerometers play a vital role in navigation and motion perception of moving objects, which covers a broad range of applications in consumer electronics, industry, medical treatment, etc. New sensing mechanisms have been pursued to improve the performance of accelerometers all the time, for instance, by synchronizing multiple resonators to achieve higher resolution [1]. The optical detection is combined with conventional micro-electro-mechanical systems (MEMS) accelerometers to achieve higher performance and multi-functionality, which merges the micro-opto-electro-mechanical systems (MOEMS) accelerometers. Optical detection benefits the accelerometers with high precision, fast response, resistance to electromagnetic interference, and the ability to work in harsh environments. MOEMS indicate the future trend of the accelerometer development [2].

MOEMS accelerometers usually rely on light-wave manipulation sensing such as intensity [3,4] and wavelength [5,6] modulations. The acceleration can be evaluated by resolving the acceleration-modulated optical signal through MEMS/NEMS spring-mass system, which is referred to as optomechanical cavity in this specific scheme. This key sensing element is a mechanically tunable optical resonator that can translate the mechanical displacement and motion in resonators into the optical signal. The investigation of MOEMS accelerometers constructed with a single optomechanical cavity is carried out widely in recent years [2]. Excellent sensitivity and resolution have been obtained by various types of MOEMS accelerometers. On the other hand, the application of the MOEMS accelerometers is limited with a small measurement range and special design that is hard for large-scale fabrication.

The sensitivity of the MOEMS accelerometers can be further optimized by making use of the mode localization phenomenon from multiple resonators coupled with each other. The first mode-localized sensing is accomplished by coupled MEMS resonators. Mode localized MEMS sensors achieve several advantages such as high sensitivity [7] and common-mode rejection [8]. From then on, a series of mode-localized sensors have been created with incredible sensitivity [9,10].

The concept of the optical mode localization is theoretically validated by [11] in coupled-resonator optical waveguides (CROWs) structure [12]. High sensitivity and common-mode rejection were achieved by evaluating the modal power ratio from resonant peaks as sensing output. Based on the four-port structure, the two output spectrum with mode localization (asymmetric mode splitting) and symmetric mode splitting allows the high-sensitivity sensing and dual-channel calibration to be carried out simultaneously, which can reduce the sensing errors [11]. Based on the same construction, we present a novel design of optical mode localization MOEMS accelerometers, which is constructed with CROWs with variable optical coupling by a spring–mass system. The proposed design achieves ultra-high sensitivity on the SOI platform, which allows the fabrication to be carried out with no extra assembly/alignment steps as required by most MOEMS accelerometers. The accelerometer is capable of achieving a large measurement range and high sensitivity in the meantime. It is the first accelerometer design that involves the optical mode localization mechanism and also a design that is capable of batch fabrication.

The design concept of the optical mode localization accelerometer is illustrated in Section 2. The accelerometer design and performance analysis are represented in Section 3. The discussion about the pro and cons of the accelerometer is carried out in Section 4. The paper is concluded in Section 5.

## 2. Optomechanical Configurations and Analysis

Mechanically tuned coupling gap (Δ*g*) and coupling length (Δ*L_c_*) in directional coupler will induce the phase shift (Δ*θ*) and the coupling coefficient change (Δ*t*). If the coupling gap and length change are caused by inertial displacement through a spring-mass system, the acceleration can be detected optically through silicon photonics ring circuitry and a suspended directional coupler. The deformation on the suspended mechanical beams (waveguides) can be assumed negligible. Substituting the suspended directional coupler into one of the bus waveguide couplings in the circuitry, the acceleration can be detected by resolving the optical resonant power change and resonant frequency/wavelength shift. The constructions of such an accelerator are illustrated in Figure 1.

The acceleration could be resolved by resonant frequency/wavelength shift from such a single ring illustrated in Figure 1. In practice, the resonant frequency/wavelength of the silicon microring resonators is sensitive to the temperature due to the thermo-optical effect. The circulated optical power in the ring resonator will heat the ring up depending on the dopant concentration of the silicon, which results in the instabilities of the sensing output. Thus, only a limited sensing accuracy of the accelerometer could be reached.

The sensing accuracy can be improved by the optical mode localization sensing mechanism. For ring circuitry, mode localization can be obtained from the coupled resonator optical waveguide system (CROW) with two rings. Regarding the short measurement range by modulating the coupling gap, the inertial displacement change in coupling length is a better choice. The possible configurations are shown in Figure 2.

As shown in Figure 2, the parametric changes are produced by acceleration on only one of the rings, which induces the asymmetrical resonant mode splitting in the output spectrum. The parametric changes in the system can be evaluated by the power ratio of the two resonant modes on the spectrum, which is referred to as mode localization sensing. By evaluating the resonant modal power ratio, resonant frequency instability can hardly affect the sensing outcome so the sensing accuracy is significantly improved. The inertial displacement on coupling length can be reflected from the mode localization of the output spectrum. High-sensitive inertial sensing can be accomplished by resolving the optical mode localization from the output spectrum. Dual-channel calibration can reduce sensing errors.

### 2.1. Parametric Change from Suspended Directional Coupler with Variable Coupling Length

The basic parametric model of a directional coupler is shown in Figure 3.

*g*_0_ and *L_c_* denote the coupling gap and length, respectively. *t* and
*κ* denotes the through and cross-coupling coefficient, respectively. *E* denotes the input electric field. The coupling coefficients *t* and *κ* can be expressed as:(1)$$t=\cos\left(\frac{L_{c}}{L_{b}}\pi \right)$$
(2)$$\kappa =\sin\left(\frac{L_{c}}{L_{b}}\pi \right)$$
where *L_c_* is the coupling length, and *L_b_* is the beat length by the symmetrical and asymmetrical modes. At the concerned wavelength *λ*, *L_b_* is calculated by:(3)$$L_{b}\left(g_{0}\right)=\frac{\lambda }{\Delta n_{12}\left(g_{0}\right)}=\frac{\lambda }{n_{1}\left(g_{0}\right)-n_{2}\left(g_{0}\right)}$$
where $\Delta n_{12}\left(g_{0}\right)$ is the index difference between the symmetrical and asymmetrical modes, and it can be adjusted by varying the coupling gap between two waveguides. $g_{0}$ represents the initial coupling gap before it is mechanically tuned. The coupling coefficients are expressed as:(4)$$t_{0}=\cos\left(\frac{n_{1}\left(g_{0}\right)-n_{2}\left(g_{0}\right)}{\lambda }L_{c}\pi \right)$$
(5)$$\kappa_{0}=\sin\left(\frac{n_{1}\left(g_{0}\right)-n_{2}\left(g_{0}\right)}{\lambda }L_{c}\pi \right)$$

The phase shift caused by the directional coupler is caused by the index difference between the super mode of the coupler and the fundamental mode of the bus waveguide. The phase shift difference between the bus waveguide and directional coupler with a given length of $L_{c}$ is solved as:(6)$$\theta_{0}=\frac{\pi L_{c}}{\lambda }\left(2n_{0}-n_{1}\left(g_{0}\right)-n_{2}\left(g_{0}\right)\right)$$
where $n_{0}$, $n_{1}$, and $n_{2}$ are the effective index of the fundamental mode (single waveguide), symmetrical and asymmetrical mode, respectively. If the phase and coupling change is caused by a variation in coupling length, the coupling and phase change are expressed as:(7)$$\Delta t^{2}=\cos^{2}\left(\frac{n_{1}\left(g_{0}\right)-n_{2}\left(g_{0}\right)}{\lambda }\left(L_{c}+\Delta L\right)\pi \right)-\cos^{2}\left(\frac{n_{1}\left(g_{0}\right)-n_{2}\left(g_{0}\right)}{\lambda }L_{c}\pi \right)$$
(8)$$\Delta \theta =\frac{\pi \Delta L_{c}}{\lambda }\left(2n_{0}-n_{1}\left(g_{0}\right)-n_{2}\left(g_{0}\right)\right)$$

To solve the Δθ and $\Delta t^{2}$ at λ = 1.55 μm, the refractive index of silicon, the waveguide width, and thickness is set at 3.48, 450 nm, and 220 nm, respectively. The phase shift change Δθ and the change in coupling coefficient $\Delta t^{2}$ at different initial coupling gap $g_{0}$ solved by COMSOL Multiphysics simulation ($n_{0},n_{1}$ and $n_{2}$) are shown in Figure 4, Figure 5 and Figure 6.

The detectable range of the variable coupling length (around 10 μm) is much wider than the one from the variable coupling gap (around 100 nm), which provides better toleration during fabrication. Large detectable $\Delta L_{c}$ also means that the large measurement range of the accelerometer is potentially achievable.

### 2.2. Optical Mode-Localized Displacement Detection with Variable Coupling Length

The parametric model of the accelerometer is illustrated in Figure 7. The resonances in the output spectrum are solved by evaluating the phase matching of the circulating power, which could be carried out by the transfer matrix [12,13] or Mason’s rule [14,15].

The power transmission from the input port to the through and drop port are expressed as:(9)$$|G_{t}{|}^{2}={\left|\frac{E_{t}}{E_{i}}\right|}^{2}$$
(10)$$|G_{d}{|}^{2}={\left|\frac{E_{d}}{E_{i}}\right|}^{2}$$

Normalizing the wavelength/frequency to θ, a typical power transmission spectrum under phase perturbation is shown in Figure 8 as an example of mode localization [11]. In Figure 8, the asymmetrical mode splitting makes the power ratio of two resonant peaks correlated to Δθ or $\Delta t^{2}$, which establishes the optical mode localization sensing.

The theoretical calculation according to the signal flow graph method can be examined and validated by the simulation result. Either the resonant power ratio ($P^{-}/P^{+}$) or the resonant phase ratio ($\theta^{-}/\theta^{+}$) can be applied as the sensing output depending on the specific sensing configuration.

We assume the coupler is based on the 450 nm wide waveguides on a 220 nm SOI wafer. Based on an initial gap of 200 to 300 nm with an initial coupling length of 15 μm, the power ratio and phase ratio according to the displacement change $\Delta L_{c}$ are shown in Figure 9, Figure 10 and Figure 11.

Calculation results are shown in Figure 9, Figure 10 and Figure 11 prove that large inertial displacement in coupling length is allowed in the design. The sensitivity and measurement range of the displacement sensing provided by $\Delta L_{c}$ is more practical compared with that by Δg (less than 1 μm). The power ratio is sensitive to propagation loss α as shown in Figure 9, Figure 10 and Figure 11. The curve will be seriously affected by the mode aliasing effect when the initial gap is larger than 300 nm. Peaks on the power ratio curve determine the sensitivity and the measurement range, which is seriously affected by α. The phase ratio would be a better choice for sensing output considering the possible fabrication uncertainty, and it provides high sensitivity as well.

## 3. Accelerometer Design and Performance Analysis

Thanks to the large available amplitude ratio, MEMS mode-localized accelerometers can achieve high sensitivity in a limited measurement range. According to calculation results in Section 2, both resonant phase ratio and power ratio can be good sensing output in our accelerometer design with the variable length configuration. If the detectable power ratio is 30 dB in practice, the available amplitude ratio can be more than 1000. The phase ratio can be more than 55 in Section 2.

To achieve the inertial displacement detection in two directions, the couplers between ring resonators should be designed with necessary misalignment as shown in Figure 12. This misalignment allows the coupling length to decrease when the proof mass moves upward and increases when the proof mass moves downward, while the regular directional couplers only decrease the coupling length.

The accelerator is based on the optical mode-localized sensing circuitry with a radius of 50 μm, waveguide width of 450 nm, and coupling gap of 250 nm on 220 nm SOI wafer, which is the same as the parameters chosen in Figure 10. The coupling length is set at 15 μm to ensure that the inertial displacement of ±10 μm is achievable. The spring–mass system is supposed to be able to work in the ±50 g range. The accelerometer design is illustrated in Figure 13. The material properties and values for simulation are shown in Table 1.

The expected inertial force corresponding to the spring design under 50 g acceleration is 10 nN, which is expected from 400 μm × 400 μm with 50% area occupied with etch holes. The mass is designed in a rectangular shape with two springs on each side connected with anchors. Each of the springs is constructed with 10 periods connected in series. The basic unit of the spring design and the displacement of the described spring-mass system under linear acceleration is shown in Figure 14. The spring constant is 28.8 N/μm (2.94 g/μm). The simulation results show good linearity.

### 3.1. Sensitivity and Measurement Range

The displacement of the spring-mass system under ±50 g is calculated to be ±17 μm by Finite Element Analysis. Regarding the mode aliasing effect and the limitation of the coupling length (15 μm), the inertial displacement allowed is in the range from −8 μm to 10 μm. For the designed spring-mass system, this range covers the acceleration from −23.5 g to 29.4 g. The optical free spectral range of the system is 4.05 nm around 1.55 μm wavelength. If we convert the inertial displacement into Δ*L* in an accelerometer configured as in Figure 10, the output of the accelerometer described in the power ratio and phase ratio are shown in Figure 15.

The number of the available power ratio is one order of magnitude larger than the amplitude ratio provided by the state-of-the-art MEMS mode-localized accelerometers. The available amplitude ratios of the reported MEMS accelerometers are shown in Table 2.

### 3.2. Resolution

The noise in a silicon photonic detection system is usually generated from LASER and photo-diode, which is related to the carriers and random numbers of photons generated. The core sensing element in this paper involves no carrier flow in silicon photonic circuitry. Based on this assumption, 1/f noise is also neglected in this analysis, so the mechanical thermo-noise is discussed only to evaluate the performance limitation by intrinsic noise from the proof mass. The thermal noise force under thermomechanical noise floor in a mechanical resonator is expressed as [24]:(11)$$F_{th}=\sqrt{\frac{4k_{B}T}{\omega_{0}^{3}mQ}}$$
where $k_{B}$, *T*, and $\omega_{0}$ denote the Boltzmann constant, Kelvin temperature, and eigenfrequency, respectively. *Q* and m denote the quality factor of the mechanical resonance and effective mass, respectively. The fundamental mode shape of the proof mass is shown in Figure 16.

According to the thermo-elastic dissipation theory, the *Q* is $4.022\times 10^{9}$ at 293.15 K calculated with $\omega_{0}=2\pi \times 352$ Hz and effective mass $m=6.9518\times 10^{-11}$ kg calculated from FEA software (COMSOL Multiphysics 5.6, Shanghai, China ). The $F_{th}$ is 3.32 fN so the proof mass is exerted with a minimum acceleration of $4.776\times 10^{-5}$ m/s^2^ (4.874 × 10 μg) with thermomechanical noise only. The resonant amplitude is 0.694 nm under thermomechanical excitation, which is negligible to the optical coupling of the suspended directional coupler. In practice, noise generated from the LASER and the photo-diode dominates the noise feature of the system, which is related to their specific design.

## 4. Discussion

The integration of the optical mode localization circuitry and dedicated proof mass design achieves high sensitivity and a large inertial detection range. The suspended directional coupler mechanically isolates the proof mass with the rest of the device so the proof mass can be designed individually without dramatically influencing the device feature. This design provides much flexibility in the proof mass design compared with a mechanical mode-localized accelerometer when considering the trade-off between the measurement range and sensitivity. However, the high output power ratio must rely on high-performance detectors and electronic peripherals with low noise and high sensitivity, which demands high fabrication costs. The output power ratio curve is sensitive to the propagating loss of the waveguide, which can be hardly tunable, so it needs to be controlled carefully during the fabrication.

The 220 nm SOI wafer is a good choice for silicon photonic devices based on the wavelength of 1550 nm. On the other hand, the thin film of silicon sets some limitations on the construction of a spring-mass system with a small footprint and large electrostatic driving force. The weak support provided by thin film also makes trouble during the structural releasing process. The working wavelength of 1550 nm requires high-resolution lithography to fabricate the silicon photonic devices with a waveguide width of 450 nm. As a result, the high fabrication cost is necessary for 1550 nm silicon photonic devices. Generally speaking, a thicker silicon layer that is compatible with longer working wavelength, for instance, MIR (middle infrared radiation), would be a future trend for MOEMS inertial devices compromising silicon photonic devices.

## 5. Conclusions

In this paper, a novel optical MEMS accelerometer that is designed on an SOI wafer with a device layer thickness of 220 nm based on the optical mode localization sensing mechanism has been proposed. The high sensitivity of the proposed design has been proved theoretically based on the suspended directional coupler and coupled ring resonators. According to the FEA simulations and numerical calculation (transfer matrix or Mason’s rule), a sensitivity of 18.9/g with a measurement range from −23.5 g to 29.4 g is obtained from the proposed device. The free spectral range is 4.05 nm around 1.55 μm. The acceleration resolution limited by thermomechanical noise is 4.874 μg. The comparison between the optical and mechanical mode-localized accelerometer demonstrated the advantages of the large available amplitude/power ratio and adjustable measurement range of the optical mode-localized accelerometer. The comprehensive performance of this design is competitive with existing MEMS mode localized accelerometers. It demonstrates the potential of the optical mode-localized inertial sensors as candidates for state-of-the-art sensors in the future. The silicon photonic circuitry isolates itself from electromechanical noises so it may contribute to a higher signal-to-noise ratio for inertial sensing. On the other hand, the proof mass occupied a large footprint of more than 400 μm × 400 μm, due to the thin device layer. Using a thicker device layer with a longer working wavelength will contribute to a smaller footprint and larger critical dimension, thus the fabrication cost and yield of the sensor could be optimized. These superiorities demonstrate that utilizing MIR circuitry in optical MEMS accelerometers may effectively improve the performance of the MOEMS accelerometers and inspire emerging applications using MOEMS technology in the future.

## Figures and Tables

**Figure 1 micromachines-14-00039-f001:**
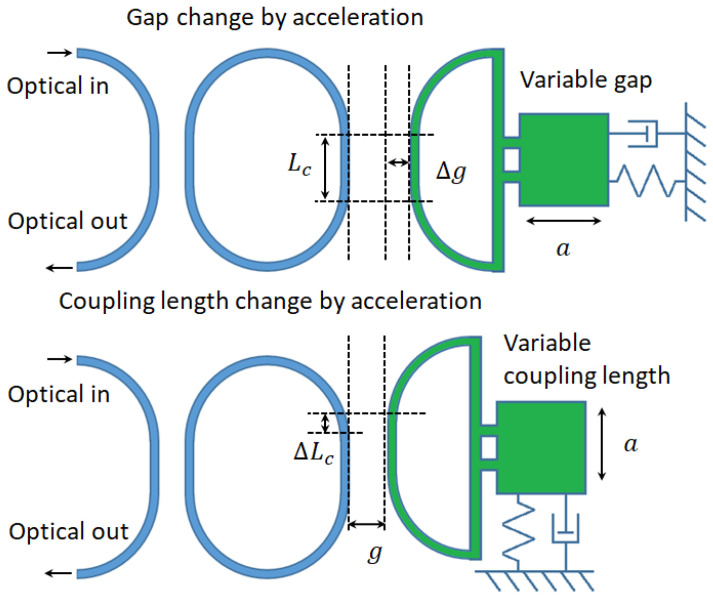
Two accelerator constructions based on a single ring resonator.

**Figure 2 micromachines-14-00039-f002:**
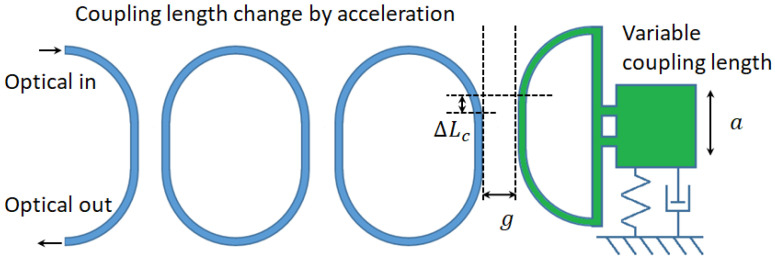
Accelerator construction based on coupled ring resonators (CROW) with variable length configuration.

**Figure 3 micromachines-14-00039-f003:**
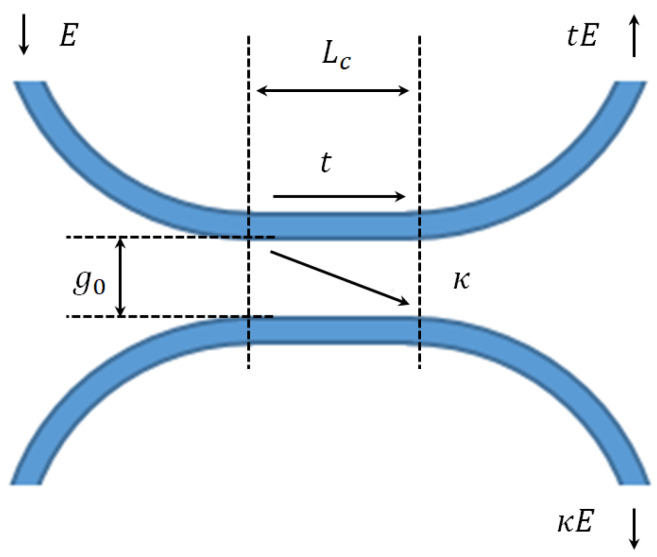
The parametrical model of the directional coupler.

**Figure 4 micromachines-14-00039-f004:**
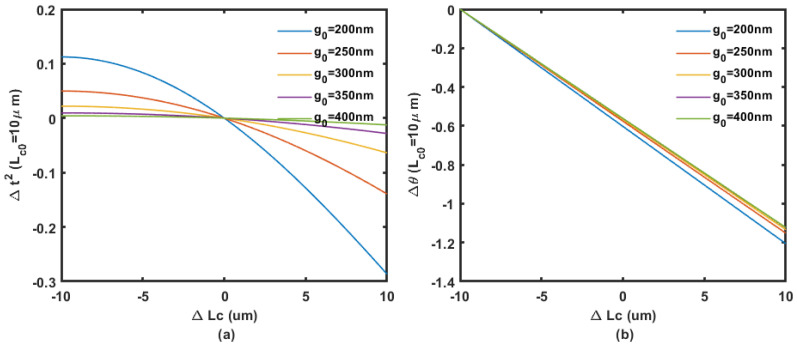
Coupling (**a**) and phase (**b**) change regarding the displacement in coupling length with 10 μm initial length at different coupling gaps from 200 nm to 400 nm.

**Figure 5 micromachines-14-00039-f005:**
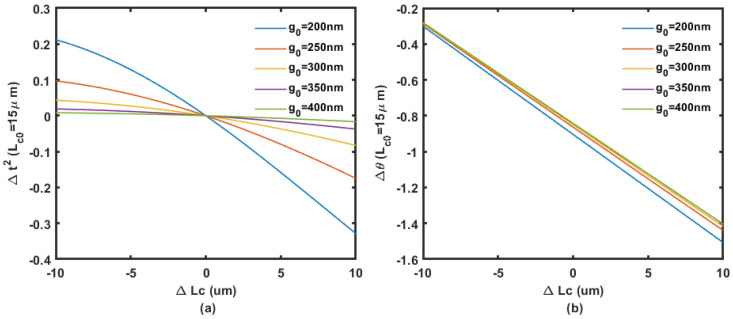
Coupling (**a**) and phase (**b**) change regarding the displacement in coupling length with 15 μm initial length at different coupling gaps from 200 nm to 400 nm.

**Figure 6 micromachines-14-00039-f006:**
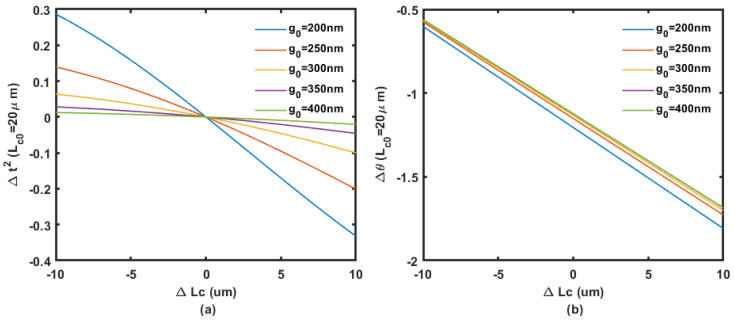
Coupling (**a**) and phase (**b**) change regarding the displacement in coupling length with 20 μm initial length at different coupling gaps from 200 nm to 400 nm.

**Figure 7 micromachines-14-00039-f007:**
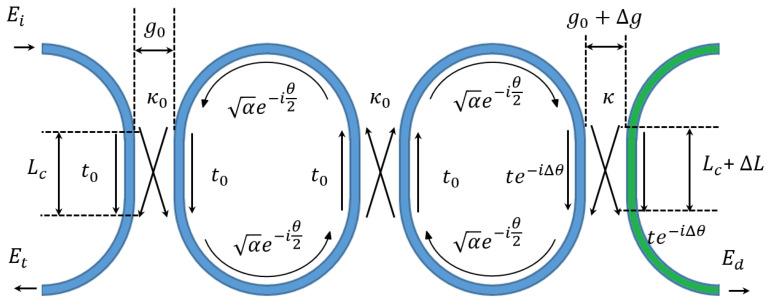
The parametric model of the accelerometer. α denotes the loss coefficient of the waveguide (when α=1, lossless). θ denotes the phase change of the waveguide mode after a round trip transmission of a ring resonator $\theta =\theta_{L}+2\theta_{0}$, where $\theta_{L}$ represents the phase shift after a round trip length without coupling, and $\theta_{0}$ represents the phase shift difference between bus waveguide and directional coupler. $E_{i}$, $E_{t}$, and $E_{d}$ denote the electrical field at the input, through and drop port of the circuitry.

**Figure 8 micromachines-14-00039-f008:**
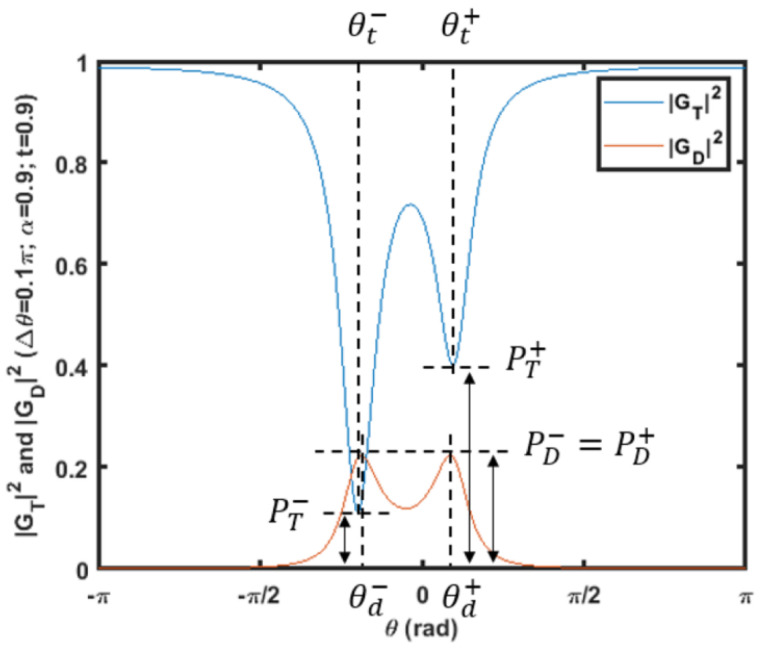
Notations for resonant peaks in $|G_{t}{|}^{2}$ and $|G_{d}{|}^{2}$ with Δ*θ* = 0.1π. The power transmission of the resonant peaks is denoted by $P^{\pm }$ to describe resonant peaks on the left (−) and right (+) hand side. $\theta^{\pm }$ denotes the phase of the resonant peaks [11].

**Figure 9 micromachines-14-00039-f009:**
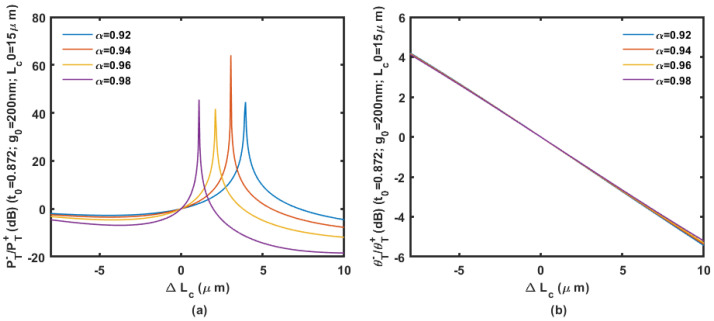
The power ratio (**a**) and phase ratio (**b**) according to the displacement change Δ*L_c_* with an initial gap of 200 nm and coupling length of 15 μm.

**Figure 10 micromachines-14-00039-f010:**
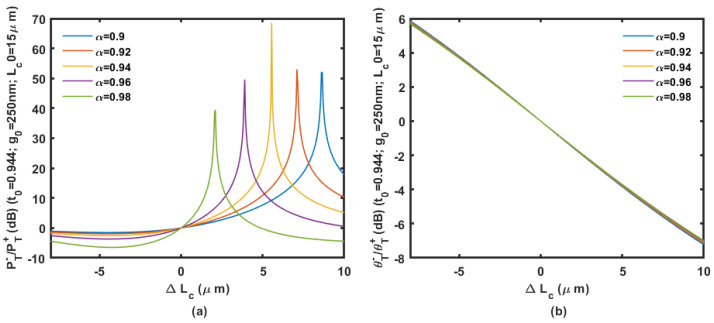
The power ratio (**a**) and phase ratio (**b**) according to the displacement change Δ*L_c_* with an initial gap of 250 nm and coupling length of 15 μm.

**Figure 11 micromachines-14-00039-f011:**
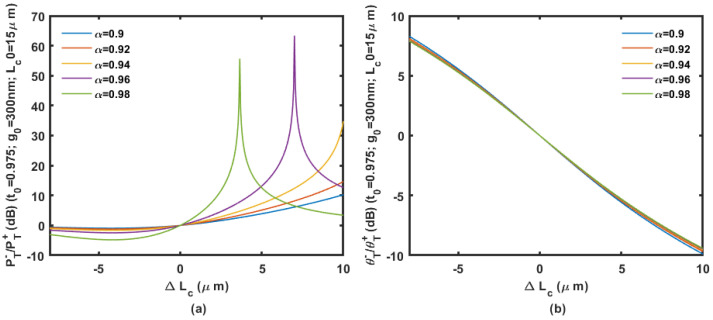
The power ratio (**a**) and phase ratio (**b**) according to the displacement change Δ*L_c_* with an initial gap of 300 nm and coupling length of 15 μm.

**Figure 12 micromachines-14-00039-f012:**
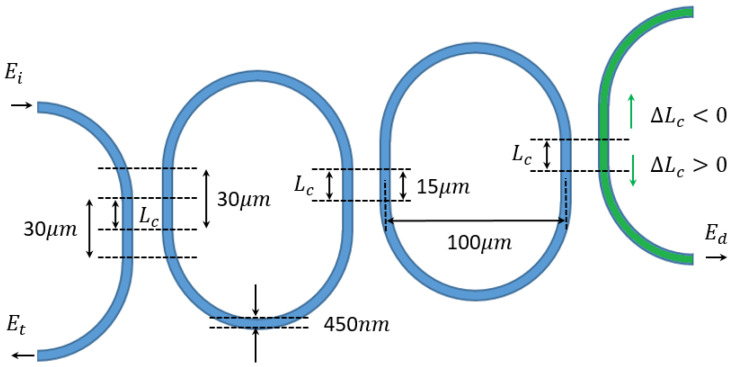
The optical circuitry design of the accelerometer.

**Figure 13 micromachines-14-00039-f013:**
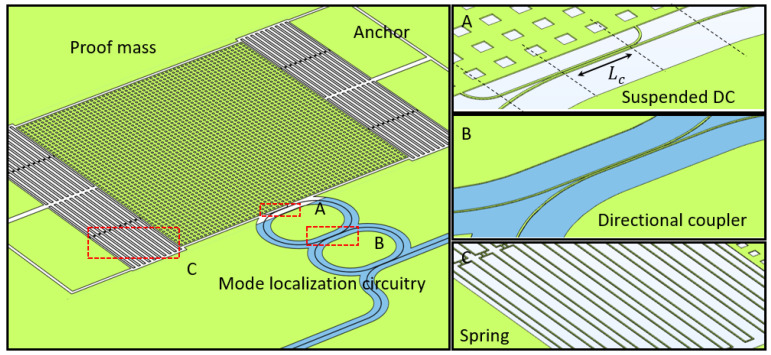
The overall design of the accelerometer.

**Figure 14 micromachines-14-00039-f014:**
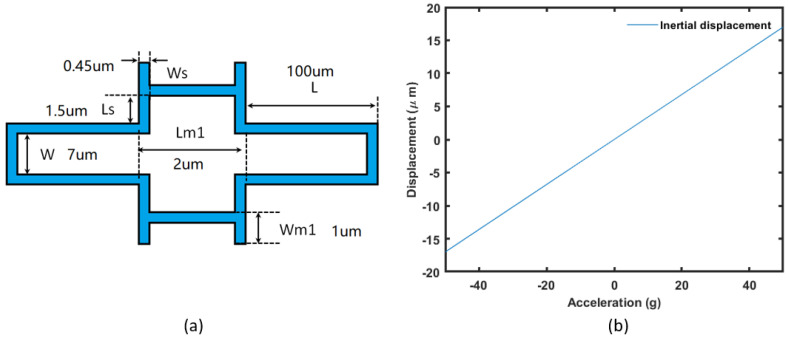
The basic unit (one period) of the spring design (**a**) and the corresponding inertial displacement (**b**).

**Figure 15 micromachines-14-00039-f015:**
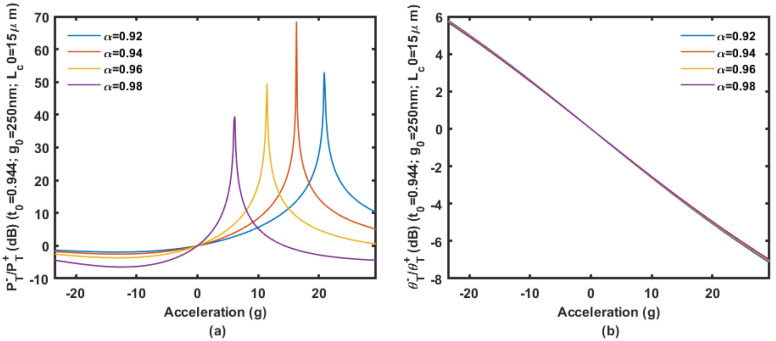
Output characteristics of the accelerometer illustrated by power ratio (**a**) and phase ratio (**b**).

**Figure 16 micromachines-14-00039-f016:**
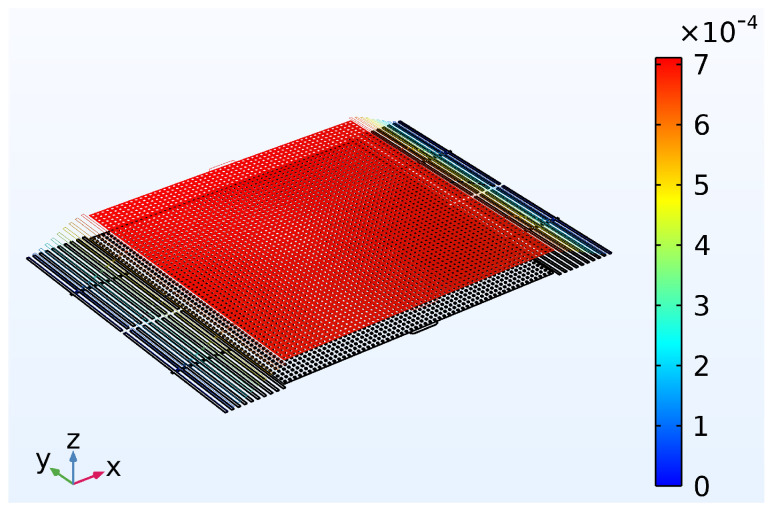
The fundamental mode of the design proof mass. The mass is vibrating in the *z*-direction only and the color bar presents the displacement in microns.

**Table 1 micromachines-14-00039-t001:** Silicon properties for simulation.

Properties	Value	Unit	Properties	Value	Unit
Thermal expansion (*α*)	2.6 × 10^−6^	1/K	Heat capacity (*C_p_*)	700	J/(kg*K)
mass density (*ρ*)	2329	kg/m^3^	Heat conductivity (*k*)	130	W/(m*K)
Young’s modulus (*E*)	170 × 10^9^	Pa	Possion’s ratio (*ν*)	0.28	1

**Table 2 micromachines-14-00039-t002:** Comparison of reported mode-localized accelerometers.

Devices	Range	Sensitivity (AR)	Available AR	Available TR
Zu et al. [16]	40 g	1.45/g	58	N.A.
Peng et al. [17]	1 g	23.37/g	23.37	N.A.
Zhang et al. [18]	0.06 g	525/g	31.5	N.A.
Pandit et al. [19]	1 g	11/g	11	N.A.
Kang et al. [20]	±1 g	4.38/g	8.76	N.A.
Kang et al. [21]	0.15 g	36.86/g	5.529	N.A.
Yang et al. [22]	0.8 g	1.01/g	0.808	N.A.
Zhang et al. [23]	±1 g	1.26/g	2.52	N.A.
this work	−23.5 g to 29.4 g	18.9/g	1000	55

N.A. means not applicable.

## Data Availability

The data presented in this study are available on request from the corresponding author.

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
