# Peer review of "Design and Simulation Study of an Optical Mode-Localized MEMS Accelerometer"

_micromachines, 2022, doi:10.3390/mi14010039_

Round 1

Reviewer 1 Report

This paper focused on the design and simulation of an optical mode-localized MEMS accelerometer.  Based on the simulation results, high sensitivity and large measurement range can be achieved by using the proposed design. I recommend for acceptance after considering the following minor modifications:

1. If possible, some experimental supports should be provided.

2. The fundings are not provided in the end of this paper.

Reviewer 2 Report

This manuscript proposes a configuration called coupled-resonator optical waveguides with two-ring resonators. The asymmetrical resonant mode splitting in the output spectrum is used for sensing, which is referred to as mode localization sensing. This configuration for sensing shows the advantage of overcoming resonance frequency instability. I have the following comments before further consideration:

(1) The authors should what are the new results and conclusions of this manuscript compared with the previously published paper (Reference [10]) from the same major authors. They look so similar. It is unacceptable for repeat publications of the same content.

(2) The caption of figure 2 is not right. It is not two accelerators, obviously.

(3) The authors say dual-channel calibration can reduce sensing errors. It is not clear. Please describe in detail how dual-channel calibration works.

(4) Why Equations (6) is not θ0 = πLc/λ*(2n0) or θ0 = πLc/λ*(-n1(g0)-n2(g0))? My intuition tells me something may be wrong. Could the authors check it and give the corresponding reference?

(5) According to equation (6), equation (8) is wrong. It should be δθ = πδLc/λ*(2n0-n1(g0)-n2(g0)).

(6) Captions of figures 4, 5, and 6 are wrong. Please check carefully. These give me a really bad impression of this manuscript sometimes.

(7) In the caption of figure 7, θ = θL+θ0, where θL and θ0 are not defined.

(8) Lines from 95 to 96 are repeated (the same with lines from 82 to 83).

(9) Line 108, “Thanks to The large” should be “Thanks to the large”; line 114, “should be design” should be “should be designed”…

Reviewer 3 Report

My general evaluation of the article titled “Design and Simulation Study of an Optical Mode-localized MEMS accelerometer” is as follows.

   It is a study in the field of "analysis of an accelerometer with a MEMS structure". It is seen that the study was organized and written in accordance with its purpose. However, this and similar studies are quite abundant in the literature. In addition, there is no fabrication process in this study. It's just a simulation run.

1. The abstract should be revised. The technique-method can be stated briefly. The purpose is not clearly stated.

2. In general, the English language of this article should be corrected. Professional help is recommended.

3.

a- Why were the physical properties of the material used in this study not given in the form of a table?

b- How did you analyze without these features? (Did you do it with COMSOL? Why didn't you specify?)

4. In the Conclusions section, the superiority and difference of the article from other existing studies should be clearly stated.

5. Make more use of recent studies in the References section. Also, increase your references.

Round 2

Reviewer 2 Report

Thank the authors for taking my suggestions to improve the manuscript. I have no other comments.

Reviewer 3 Report

Many thanks to the authors for their hard work. This article is a simulation study only. There are many similar studies in the literature. I think that experimental data should also be added to the study.